# Durability and Degradation of PVC-P Roofing Membrane—Example of Dynamic Fatigue Testing

**DOI:** 10.3390/polym14071312

**Published:** 2022-03-24

**Authors:** Andrej Ivanič, Samo Lubej

**Affiliations:** Faculty of Civil Engineering, Transportation Engineering and Architecture, University of Maribor, Smetanova ulica 17, 2000 Maribor, Slovenia; fgpa@um.si

**Keywords:** environmental impact of polymers, durability, polymer degradation

## Abstract

This paper presents a study of PVC-P waterproofing membrane Specimens. The Specimens were taken from different segments of a flat roof after a service life of 11 years. The reason for analysing the condition of the Specimens was the apparent degradation of the waterproofing, which no longer guaranteed the watertightness of the roof. The analysis of the performance of the Specimens was based on the control of the mechanical properties, which were compared with the declared values. The mechanical properties of the degraded PVC-P waterproofing membranes with a polyester mesh backing, which are prescribed by the standards, do not usually deviate from the declared properties. This often poses a problem for liability and warranty claims due to the poor quality of the waterproofing membrane. There may be several causes of degradation of PVC-P. For this reason, in this paper, we present the possibility of controlling the properties of PVC-P waterproofing membranes using cyclic dynamic fatigue, microstructure analysis using scanning electron microscopy (SEM) and energy dispersive spectroscopy (EDS). The results show that the cause of the deterioration of the PVC-P waterproofing membranes is often related to dehydrochlorination of the polymer. However, the deterioration of the mechanical properties of the PVC-P waterproofing membranes can be successfully demonstrated by cyclic dynamic fatigue.

## 1. Introduction

The global production of PVC continues to grow, which is now also evident in the construction industry. PVC-P-based waterproofing materials compete very successfully with bitumen-based waterproofing materials in the construction of flat roofs. The reason for the exponential growth in the use of PVC-P roofing materials in the construction industry is mainly that PVC-P roofing materials are installed in a single layer as the final waterproofing layer of flat roofs. For bitumen-based waterproofing, the rule of thumb is that it should be installed in at least two layers as the final layer of flat roofs. The advantage of PVC-P over bitumen roofing in the construction of flat roofs for the above reasons lies not only in the amount of material used but also in the amount of work involved. This, of course, also means a financial advantage.

Polyvinyl chloride (PVC) is a widely used polymer, not only in industry, but also in daily life. PVC is a material that can be applied in many different fields, such as building and construction, healthcare, and electronics [1]. The history of PVC started during early 1870, when the first polymer was obtained by the polymerization of vinyl chloride. However, the material obtained from the polymerization of vinyl chloride was stiff and brittle. Hence, it was not ideal for industrial production. By 1926, American chemists discovered how to plasticize PVC and, since then, many PVC-based products have been commercialized [2,3]. Nowadays, PVC is used in many different fields, such as in the construction industry (pipes, windows, carpet, plumbing, etc.), the electrical and electronic industry (instrument components, housing, sheaths for cables and wires, etc.), the automotive industry, in food packaging, in medical equipment and many others [4]. PVC is one of the most commonly used thermoplastic materials today in worldwide polymers. PVC has low cost, high performance and high possibility of producing a variety of products from different processing conditions and techniques (Norazlina et al. [5]). PVC is extremely important, with useful molecular structure and morphology in every area; for example, it is used as a thermoplastic due to its many valuable properties, such as low price, good process ability, chemical resistance and low flammability (Kok et al. [6]).

When analysing the quality control of flat roofs, complaints have been very common in the construction industry over the last 10 years. The cause of the complaints is usually not only in the execution defects of the waterproofing contractors, but very often also in the questionable quality of the PVC-P waterproofing membrane. The guaranteed certificate is usually issued by the roofing manufacturer, who is also obliged to ensure the correct installation of the insulation materials, in accordance with the regulations for the performance of waterproofing works. Problems relating to the duration of the guarantee period very often arise already at the stage of obtaining the guarantee certificate. In construction practice today, there is an unwritten rule that, if it is not possible to obtain a 10-year guarantee from the manufacturer of the insulation material, then the chosen material will not survive this period without degradation or damage. Of course, this applies mainly to materials in the lower price range, where the risk is greatest. A direct proportionality applies—the lower the price, the shorter the lifetime [7]. This is very often the case for materials containing recycled PVC, which also results in a significant price reduction due to the reduced carbon footprint.

Physical properties of all roof systems change with age and outdoor exposure. The change in physical properties of a roof membrane may be the result of many factors. A few factors that may affect the physical properties of a vinyl membrane include chemical formulation stability, thickness of the polymer, reinforcement, method of manufacturing, geographic location, heat and ultraviolet radiation exposure, other products used in conjunction with the membrane, and roof slope. These factors cannot adequately be simulated in any test program. The certainty of service life predictions increases with increasing application experience [8].

PVC is a very useful type of plastic with a very wide range of applications. PVC is a polymer in which more than half of the content by weight consists of chlorine [9]. PVC is produced by polymerization of the vinyl chloride monomer. PVC comes in two basic forms: rigid (sometimes abbreviated as RPVC) and flexible. The quality of the material is very important in roofing, and it must have the properties to maintain its declared properties over a long service life. PVC is known to be susceptible to the ageing process. In the case of the ageing of waterproofing materials, several mechanisms have been identified that can cause degradation and associated premature failure of PVC-P waterproofing materials. The position of the polymeric waterproofing membrane in the flat roof system is a very important consideration. In inverted flat roofs, the thermal insulation is placed above the waterproofing membrane and is usually separated by a layer of polyester felt. Such a construction solution always involves the use of polymeric insulation materials. The thermal insulation layer in such a construction is always above the waterproofing layer and is extruded polystyrene (XPS). The PVC-P waterproofing membrane in inverted flat roofs is always loaded with different building materials. There is no alternative material for the thermal insulation in such a design. Insulation materials in inverted roof construction are always polymer-modified materials. It is known that some polymeric materials exhibit interactions with each other that can affect their durability and stability, which is due to their incompatibility with each other [10]. In the case of direct contact between a PVC-P membrane and XPS thermal insulation in a flat roof system, the migration of plasticizers (plasticisers) is triggered due to the influence of temperature during the summer period. Heat is an important factor in the weight loss of the PVC-P waterproofing membrane. The loss of mass due to plasticizer migration causes changes in the volume of the material and high internal stresses in the material. This is seen in the worst case on the PVC-P waterproofing layer as membrane rupture, wrinkling and anomalies, at poorly made welding or adhesive interfaces. Thinning of the already thin waterproofing membrane can lead to a situation where the curing phase (polyester mesh) is no longer bonded well to the matrix, which is PVC. One main advantage of PVC sheets is that the entire roof membrane can be joined by welding the joints with solvent or with air heated to 425 °C. This membrane can also be welded to metal flashing that has been factory-coated with PVC. The result is a continuous roofing assembly. PVC sheets remain flexible at temperatures as low as −40 °C. They are ideal for re-roofing and repairs, because of their high permeability. Moreover, white reflective vinyl membranes contribute to reducing urban heat island effects, can be produced in a wide spectrum of colours to meet desired aesthetic features of building, have high resistance to puncture and impact and have excellent resistance to flame exposure and subsequent fire propagation [11]. Loss of plasticizers is a concern with certain PVC roofing products [12,13] because it causes embrittlement in the PVC sheets. This can be controlled by using high-molecular-weight plasticizers that have less of a tendency to volatilize or migrate out of PVC resin. PVC-P membranes have good resistance to industrial pollutants, bacterial growth, and extreme weather conditions. Minor damage to the sheet during installation or in service can be easily repaired by patching the hole using heat or solvent. PVC is incompatible with bituminous materials; therefore, care must be taken to avoid direct contact with bitumen [11].

Flat roofs are very often constructed as unvented roofs. Such constructions consist of a vapour barrier on a load-bearing substrate, a layer of thermal insulation above the vapour barrier and a waterproofing membrane. The waterproofing in such a construction may be made of different materials; if synthetic polymeric membranes are used, then the construction is classified as flat roof, where the waterproofing is mechanically fixed to the supporting substrate by telescopic screws or weighted with different building materials, such as gravel or similar. The separation layer between the waterproofing and the weight is usually made of polyester felt.

The development of synthetic polymer membranes, especially PVC-based systems, has increased the number of pathological changes in these membranes. In particular, the high competition and the downward trend in price are also causing manufacturers to use low-quality and incompatible raw materials. A characteristic feature of such products is that they may retain only some of their basic properties during the exfoliation period, which is not the case for inertness to ageing. The ageing effect of PVC always results in the loss of some essential quality characteristics [14]. That direct contact of PVC-P membranes with extruded polystyrene (XPS) causes plasticizer migration, dehydrochlorination and oxidative degeneration, which have been described by researchers [15,16]. The incompatibility of expanded polystyrene (EPS) thermal insulation and PVC-P waterproofing is well known [16]. Fang et al. [11] carried out a study on the morphology of poly(vinylchloride)/polystyrene blends by the electron microprobe analysis method. They concluded through their results that PVC and PS are incompatible. The problem of degradation of PVC-P due to possible direct contact in the structure is also pointed out by manufacturers of synthetic waterproofing membranes in their technical instructions. The problem of the installation of PVC-P roof waterproofing in a structure with EPS thermal insulation should be solved by the installation of a polyester felt separation layer.

The effect of dehydrochlorination of PVC is also strongly related to the degradation of PVC-P waterproofing membranes. It is known that PVC undergoes a process of dehydrochlorination or elimination of PVC. This process produces compact double bonds on the polymer chain, but it is not yet fully confirmed whether the resulting chloride remains in the polymer medium or volatilises as HCl. It has been confirmed that PVC-based materials undergoing an accelerated dehydrochlorination process lose good mechanical properties [17].

The mechanical properties of PVC-P roofing membranes are specified by European Standards (EN) in Europe and by ASTM in North America. The most common mechanical property tests specified in the Technical Specification for PVC-P membranes are tensile strength and elongation at break. Both properties should be measured in the longitudinal and transverse direction of the roll. The change in mechanical properties may be due to several factors such as stability of the chemical formulation, thickness of the membrane, type of reinforcement in the membrane, exposure to temperature, ultraviolet radiation, compatibility with materials with which they are in direct contact [18]. These factors cannot be accurately simulated in the laboratory; therefore, it is very important to obtain comparative data by field sampling when analysing the causes of the change in mechanical properties [19]. Synthetic membranes for waterproofing are composite materials. They consist of a matrix, which is a polymer, and a reinforcement, which can be a polyester mesh or glass fibre. It is characteristic of PVC-P roofing membranes that the tensile strength of a membrane reinforced with polyester mesh depends mainly on the strength of the reinforcement, but if the reinforcement is glass fibre, the strength also depends on the strength of the polymer matrix. PVC-P roofing membranes containing a glass fibre reinforcement phase are usually reinforced with discontinuous fibres which are not woven. The same applies to PVC-P membranes without reinforcement.

Due to the lack of specific studies on the durability of PVC-P membranes for roofing applications, it is very important to include, in the present research, selected studies that address the research already carried out on PVC materials, as the similarity of the mechanisms sheds light on the degradation mechanisms [20].

Hydrophilicity is an important property of PVC-P waterproofing. It is known that this property is assessed by measuring the contact angles of the sessile droplets. The general observation is that the water repellence of PVC-P membranes on aged roof membranes decreases. Increasing the hydrophilicity of PVC-P membranes increases undesirable water permeability and may cause damage to other roof layers [21,22].

## 2. Materials, Characterization and Methods

### 2.1. Specimens from the Case Study

The flat-roofed building investigated in this paper has a large volume and is composed of several segments, roughly formed by four bays. The architecture of the building is typical of buildings intended for education. The roof design of the building is a flat, semi-pitched roof, which is accessed via special exits. The roof area of approximately 20,000 m^2^ also contains mechanical rooms and light rooms, which are of the prefabricated type with walls made of façade panels. The roofs of the building segments listed above are not identical in composition. The flat roofs of all four sections are solid, unvented roofs, while the flat roofs of the roofed buildings are light, flat, unvented roofs. The solid roofs of all four wings of the building are constructed as a system of unvented flat roofs on pitched (lightweight) concrete, with vapour barrier, thermal insulation, polyester felt separating layer, waterproofing, polyester felt separating layer and gravel weighting. The load-bearing structure of this roof is an AB panel. The buildings located on the solid roof shall have a flat, unvented roof system on trapezoidal sheeting, with a vapour barrier, thermal insulation, a polyester felt separating layer and waterproofing. The supporting structure of these roofs is metal.

The first approach on which the experimental design was based was a visual inspection of the roof and its water retention status. Based on this inspection, we were able to form a general impression of the quality of the works carried out, relating to the roofing and carpentry works and the state of the waterproofing of the roofs of the building. Our conclusion, based on the engineering assessment, is that the roof is leaking and that this leaking causes extensive damage to the building after each period of rain. Roof waterlogging causes damage to the rooms directly under the roof. The maintenance service has buckets permanently placed in the areas where the roof is constantly clogged, which are mainly at the penetrations of the storm water drainage system, these are emptied as necessary.

The visual inspection revealed several construction defects resulting from non-compliance with the rules of the roofing and carpentry trade and obvious signs of degradation of the waterproofing, which is a single-layer synthetic membrane made of PVC-P. The findings of the visual inspection are that the waterproofing is crumpled and stretched. The shrinkage of the waterproofing and the improper execution of the finishing on the roof attic led to damage and consequently to waterlogging. Many leaks are also caused by inadequately executed carpentry finishes.

The second phase of the approach was to detect the composition of the roofs and to Specimen the materials incorporated in the roofs. The materials were Specimend at random locations on the flat roofs. Figure 1 shows the consequences of inadequately executed waterproofing finishes. Figure 1a shows damage to the detail of the waterproofing finish—tearing out of the finish below the façade attic at the fixing point due to shrinkage. Figure 1b shows the waterproofing wrinkling phenomenon, and Figure 1c shows the waterproofing membrane stretched and no longer vertically terminated at the attic, also due to shrinkage of the waterproofing.

Due to the obvious signs of degradation of the PVC-P waterproofing, the sampling of the waterproofing and the research work prior to the sampling was defined by a research work plan, which included:PVC-P membrane sampling in the field;Preparation of Specimens for mechanical tests, which were planned to be carried out using standard tensile tests and non-standard dynamic tensile testing;Preparation of Specimens for the surface condition inspection of the PVC-P membrane—macrostructure determination and thickness measurements;Preparation of Specimens for microstructure determination and chemical analysis.

### 2.2. Sampling and Specimen Preparation

The sampling location chosen at the roof attic, where the waterproofing transitions from horizontal to vertical, was chosen on purpose. Two Specimens were obtained at this roof location, of which the vertically laid part of the waterproofing was exposed to insolation, but the horizontal part was not, as it was covered with polyester felt and a layer of gravel between installation and sampling. Unfortunately, the termination of the transition of the waterproofing from the horizontal to the vertical direction was not carried out according to the rules of the roofing and carpentry trade because of the lack of mechanical fixing at this transition. Additionally, the vertical termination did not have a separating layer of polyester felt between the thermal insulation layer on the attic and the waterproofing. The contractor also failed to cut off the excess waterproofing at the vertical transition at this point. For this reason, at this location, a Specimen of the waterproofing, which was not exposed to the insolation, was also obtained. The sampling location is shown in Figure 2a. The waterproofing pieces Specimend were prepared for mechanical testing and microscopy.

Figure 2b shows a Specimen of all three membranes prepared for optical microscope thickness measurements. The PVC-P membrane pieces were first cleaned of impurities with 96% ethanol and the cleaned pieces were embedded in Bakelite, the prepared Specimen was finely polished with an abrasive medium before microscopy.

Specimens were taken at the facility, which we have labelled in this article as:Specimen 1—PVC-P membrane in contact with EPS and exposed to insolation;Specimen 2—PVC-P membrane in contact with the EPS and not exposed to insolation;Specimen 3—PVC-P membrane separated from the EPS layer by a layer of polyester felt and not exposed to insolation.

It is also clear from the description of the sampling site that Specimen 1 and 2 were also significantly more temperature-stressed than Specimen 3. The PVC-P membrane was installed in the roof structure of the building 11 years ago.

### 2.3. Mechanical Testing

In the laboratory, Specimens were prepared for mechanical testing according to EN 12311-2 [23]. According to this standard, the tensile strength and elongation at break of the Specimen were determined on the prepared Specimens. This test is the static test, as declared by the standard.

A non-standard dynamic test was also carried out on the three Specimens by cyclic tensile loading of the Specimen in the elastic range. Each Specimen was subjected to a cyclic tensile test with 150,000 cycles of a sinusoidal shape in a load range between 300 and 400 N, at a frequency of 1 Hz. After 150,000 cyclic loads, the Specimens were tensile-loaded to rupture, according to EN 12311-2.

Mechanical testing was carried out using a Zwick/Roell Z1010 universal hydraulic machine (Zwick/Roell, Ulm, Germany) with a capacity of 10,000 N. An associated optical extensometer (Zwick/Roell, Ulm, Germany) with an accuracy of 5 μ was used to determine the specific strains in the static tensile tests. For the dynamic tensile tests, the specific strains were measured by means of a cross-head displacement measurement. Zwick TestXpert III software (version 1.6, Zwick/Roell, Ulm, Germany) was used to process the results of the mechanical tests.

### 2.4. Examination with a Stereo Optical Microscope

The surface condition of the collected PVC-P membrane Specimens was checked using an OLYMPUS SZX10 optical stereo microscope with the associated AnalySIS Auto hardware (Olympus Corporation, Tokyo, Japan). The thicknesses of the Specimens were also measured on a specially prepared Specimen containing pieces of all three PVC-P membranes Specimend, which were embedded in a Bakelite paste and the surfaces finely polished.

### 2.5. Microstructure Determination and Chemical Analysis by Electron Line Microscopy

The microstructure of the collected PVC-P membrane Specimens was determined using a FEI SEM SIRION 400 NC high resolution field emission scanning electron microscope (FEI Company, Hillsboro, USA). The microscope is equipped for microchemical analysis with an EDS Oxford INCA 350 energy dispersive spectrometer (Oxford Instruments, High Wycombe Bucks, England), which allows qualitative and quantitative microchemical analysis in point, line and by area.

## 3. Results and Discussion

### 3.1. Results of Mechanical Testing

#### 3.1.1. Static Tensile Test

The waterproofing Specimens taken from the roof are made of 1.5 mm thick PVC-P with polypropylene mesh reinforcement. The declared values of the mechanical properties given by the manufacturer and determined according to EN 12311-2 are:Tensile strength ≥ 1100 N/5 cm;Elongation at break ≥ 15%.

The static tensile test according to EN 12311-2 was carried out on three Specimens for each batch of Specimens from the sampling points described. The results are the working diagrams—representative curves—shown in Figure 3, and the individual values are shown in Table 1.

The mechanical properties of PVC-P waterproofing membranes depend on the properties of the polymer, which is PVC, the mechanical properties of the curing phase, which is polyester mesh, and the proportion of filler used. The mechanical properties are further influenced by the distribution of the filler and the curing phase over the polymer matrix, including the adhesion between the matrix and the curing phase. Figure 3 shows the stress/strain diagrams giving the basic mechanical properties of the Specimens tested. The shape of these curves is typical of plastic polymers. The beginning of the curve from the coordinate origin to the first bend represents the elastic region of the linear plastic deformation range, followed by the transition from elastic to plastic deformation, in which region the maximum stress usually occurs. For polymer materials, this peak stress is identified as the yield stress.

The tensile force then causes very large plastic deformations until the material fractures. The value of the stress at which fracture occurs is defined as the tensile strength for polymeric materials. This form of stress/strain curves is typical for viscoelastic materials [24]. The curve for Specimen 1 has the steepest slope. It should be noted that it is the only Specimen that has been subjected to insolation and direct contact with EPS. The mechanical behaviour of polymeric materials changes with ageing. Plastomers subjected to the ageing process are characterised by a significant increase in the slope of the curve in the liner yield range. This is due to an increase in the stiffness of the material as the material hardens, but a decrease in elongation at break, which averages 15.82%. The decrease in elongation at break is due to microstructural changes associated with ageing of the material, which are usually due to the degradation of the polymer chains [25]. The elongation at break is a very important mechanical quantity. Microstructural changes are also associated with reduced ductility of the material [26,27]. Khemici et al., in their studies concerning the temperature ageing of PVC between 39.5 and 80 °C, state that the modulus of elasticity increases with ageing time. A higher modulus of elasticity of a material means a lower elasticity of the material, which they attribute to the effect of a decrease in molecular mobility associated with structural relaxation, accompanied by a decrease in the free volume in the material [28]. Our measurements of the secant modulus at a force of 350 N confirm this, as we measured a modulus of 3.55 MPa for Specimen 1, 1.12 MPa for Specimen 2 and 3.43 MPa for Specimen 3.

Comparing the curves for Specimen 2 with those for Specimen 1 and 3 in Figure 3, it is clear that Specimen 2 has retained a certain degree of elasticity as the elongation at break for these Specimens is on average 65.19%, for Specimen 3, the average is 22.6%, and for Specimen 1, the average is at least 15.82%. Specimens 2 and 3 were not subjected to the same temperature stress as Specimen 1 between installation and test because they were protected by EPS thermal insulation. The measured tensile breaking forces of the Specimens are higher than the declared values. The average force measured on Specimen 1 was 1558.49 N, on Specimen 2 it was 1433.96 N and on Specimen 3 it was 1419.85 N. The results obtained show that the strength and stiffness of the Specimens increase due to thermal ageing but their overall deformation capacity decreases, which is consistent with the findings of Garrido M. et al. [20].

#### 3.1.2. Dynamic Tensile Test

The results of the dynamic tensile test for the three Specimens considered are shown by the working diagrams in Figure 4 and the calculated values of the mechanical quantities in Table 2.

In the working diagrams, the curves are bounded by regions A, B and C.

Region A graphically represents the conditioning of the Specimen with 150,000 loading cycles with a force of 300–400 N and a frequency of 1 Hz. It is not possible to graphically represent such many cycles on a diagram; therefore, the working diagram shows the change in force (ΔF) on the ordinate axis and the change in specific strain (εA) on the abscissa axis. A typical segment from a cycling diagram is shown in Figure 5. Region B shows the stretching of the Specimen to the maximum force. Region C shows the behaviour of the Specimen after the maximum force has been reached until the Specimen breaks.

The load-bearing capacity of polymer materials depends not only on the material properties but also on the type of mechanical load. If a material is loaded dynamically, then it behaves differently from a static load. The curves shown in Figure 4 give the force versus specific strain relationship for a tensile test with a cyclic load of 150,000 cycles. The measured mechanical characteristics are given in Table 2 for each individual Specimen. Region A—representing a load of up to 400 N with a tensile testing head feed rate of 200 mm/min and then cyclic loading in the range 300/400 N at a frequency of 1 Hz—is determined by measuring the specific strain ε_A_. The ε_A_ values for Specimen 2 and 3 are 14.28% and 17.02%, respectively. After the cyclic loading, each Specimen was tensile tested to fracture. Again, the tensile testing head feed rate was 200 mm/min. The specific strain at maximum force, ε_B_, and the specific strain at break, ε_C_, of the Specimen were measured. The value of ε_B_ is the highest for Specimen 1 at 40.88%, for Specimen 2 and 3 it is 27.75% and 29.38%, respectively. The value of ε_B_ for Specimen 1, which was dynamically loaded, increased by 263.6% compared with the average value measured in the static test. ε_B_ was the highest for Specimen 2 and 3 at 40.88%, for Specimen 2 and 3 it was 27.75% and 29.38%, respectively. The values of the specific strains ε_B_ and ε_C_ for the dynamic test of Specimen 2 and 3 are comparable to the average values of the specific strains at maximum force (ε_Fmax_) and at rupture (ε_Break_) measured in the static test. Again, the comparison of the specific strains of the Specimens subjected to higher temperature loading shows that the deformation capacity is reduced [20]. The comparison of the breaking forces shows that the breaking force in the dynamic test is reduced rigorously for Specimen 1 compared with the static tensile test. The breaking force for Specimen 1 in the dynamic test is 37.96% lower than the average value of the breaking force in the static test. This value is reduced by 11.08% for Specimen 2 and by 22.51% for Specimen 3. The behaviour of the PVC-P roof membrane Specimens in dynamic tension is not comparable to the results of the static tensile test. For all three Specimens it was observed that the Specimen breaks brittlely after the dynamic tensile test, only Specimen 2 shows a neck-narrowing phenomenon in the fracture area. There is a colour change on Specimens 1 and 2 exposed to UV radiation—see Figure 6.

### 3.2. Results of the Stereo Optical Microscope Examination

Figure 7 shows the surface of Specimen 1. On both the upper and lower surfaces, we observed degradations which are fistula-like formations. These degradations always occur at the crossing points of the curing phase—the fibres of the polyester mesh. The characteristic area of the degraded site on the upper side exposed to insolation is about 1,800,000 μm^2^. The thickness of the Specimen was measured at two characteristic locations along the fibre mesh and in the cross-section of the composite with the polyester mesh fibres—see Figure 8.

Figure 9 shows the surface of Specimen 2; interestingly, the surface of this Specimen is more diffuse on the upper side than that of Specimen 1. The surface of this Specimen also shows no noticeable degradation similar to the fistula-like formations already mentioned for Specimen 1. The thickness measurement of Specimen 2 is given in Figure 10. It can be seen that the thinning of the thickness of this Specimen is much less than that of Specimen 1, and the variation in thickness at the different measurement points is not noticeable.

Figure 11 shows the degraded surface of Specimen 3. This surface also shows degradation similar to that of Specimen 1. The spherical degraded areas appearing at the intersections of the polyester mesh filaments are very visible on the upper side, see Figure 11a, while they are barely visible on the lower side, see Figure 11b. The areas of spherical degradation on the upper side of the membrane are different and vary between 700,000 μm^2^ and 1,200,000 μm^2^, they are significantly smaller than those observed for Specimen 1. Additionally, when measuring the thickness of this Specimen, we measured the thinning of the Specimen—the measurements are shown in Figure 12.

The thicknesses of the Specimens measured with the optical microscope are given in Table 3. The optical microscopy performed on the PVC-P roof membrane Specimens showed that the thickness thinned the most on Specimen 1 and 3 and the least on Specimen 2.

The surface damage observed in Specimens 1 and 2 is due to microstructural changes in the material and reduces the elongation at break [25]. Under natural conditions, solar irradiation (UV irradiation [2]) is the main factor in the deterioration of waterproofing membranes. The deterioration of polymers involves changes in both their chemical structure and their physical properties. These changes usually deteriorate the original properties of the material. Polymer degradation also includes biodegradation, pyrolysis, oxidation, mechanical and photocatalytic degradation. Polymers are sensitive to adverse environmental influences. Oxygen and its active forms, humidity and atmospheric pollution by nitrous oxide, sulphur dioxide and the ozone affect the long-term properties of polymers. Physical processes, such as thermal contraction, thermal expansion, various forms of mechanical stress and UV radiation also have an important influence on the properties [29]. Some of the causes of degradation of PVC-P membranes, such as high temperature and UV radiation, are responsible for two main degradation processes of polymeric materials, such as colour change [30] and dehydrochlorination, in which the element Cl is released in the form of hydrogen chloride (HCl) [31].

As can be seen in Table 3, Specimen 1 and 3 have shrunk compared with Specimen 2. This is due to the temperature ageing to which they would have been subjected, resulting in the migration of plasticizers and negative volume changes. Studies by Dunn et al. [32] and Audoin et al. [33] have indicated that the reduction in elongation at break of the Specimen in tensile testing is also due to the migration of the plasticiser from the PVC.

### 3.3. Results of Surface Examination, Microstructure Determination and Chemical Analysis of EDS

From the Specimens taken from the roof, special Specimens were prepared for electron microscopy, which were pre-sputtered with gold. A systematic examination of the microstructure of the top layer of the PVC-P waterproofing membrane was carried out on each Specimen. The results of the examination are shown by SEM-SEI photographs of the area, a typical spectrum of the planar EDS analysis and a table summarising the results of the EDS analyses. The microchemical analysis of the investigated areas is given in weight percentages for the individual chemical elements present.

Specimen 1 has a highly degraded surface on the upper surface with many cracks, some areas (yellow arrows) behind the labile with poorly bonded surfaces, and in some places the polyester mesh is no longer covered by the PVC-P (blue arrow) layer. An example of such an area of microstructure is shown in Figure 13.

Figure 14 shows the area over which the EDS plot analysis was performed (a) and the EDS spectrum (b).

Figure 15a shows an area with a pronounced degradation of the PVC top layer, to the extent that the polyester mesh is no longer completely covered by PVC. The left-hand side (b) shows a typical EDS spectrum of this area. Figure 16 shows the 5 areas where the EDS analysis was carried out. Table 4 summarises the EDS analysis results for Specimen 1.

Figure 17 shows the typical microstructure of the upper surface of Specimen 2. Degradation is also evident on this surface, with microcracks (yellow arrows) predominating, but less extensive than Specimen 1.

Figure 18a shows the EDS analysis area covering a typical area interspersed with microcracks, on the right (b) is the EDS spectrum of this area.

Figure 19a shows a part of the area with a slightly more pronounced degradation in the form of a flake which is almost separated from the base material—the PVC-P matrix. The EDS spectrum for this area is shown on the left (b).

Figure 20 shows the characteristic area of the upper side of Specimen 2 with the location of the five EDS analysis areas. Table 5 summarises the EDS analysis results for Specimen 2.

The characteristic microstructure of the upper surface of Specimen 3 is given in Figure 21. This surface is also highly degraded behind many microcracks (yellow arrow) and gaps appearing at the crossing points of the polyester mesh (red arrow). Figure 22 and Figure 23 show such a structure with the area of the EDS spectrum marked (a) and the EDS spectral analysis plot (b). Figure 24 shows the area of the five surfaces on which EDS analysis was performed and Table 6 gives a summary of the spectral analysis results for Specimen 3.

PVC has been one of the most important technical polymers for many years. PVC is known to cleave to hydrogen chloride at high temperatures, forming polyene sequences and discolouring the polymer. Up to about 220 STC, hydrogen chloride is the only volatile decomposition product. In the presence of oxygen, in addition to dehydrochlorination, oxidation reactions can also take place, which can also trigger chain scission. It is quite clear that the decolourisation during thermal degradation of PVC is related to the formation of a sequence of conjugated double bonds in the polymer chains. The colour becomes more intense as the HCl is cleaved, but the exact quantitative relationship between the colour and the amount of HCl cleaved is not yet known [34]. It is known that high temperature and UV radiation can cause degradation of PVC-P membranes. This degradation can be defined by two main degradation processes, which are polymer discolouration and dehydrochlorination, which are interrelated [30,31], and loss of plasticizer [10,33]. The dehydrochlorination of poly(vinyl chloride) has been the subject of much investigation, particularly with the view of developing greater stability in PVC polymers and copolymers. Like many polymeric reactions, dehydrochlorination is a complex process. The vinylene groups, created by the elimination of HCl from adjacent carbon atoms in the following chain:



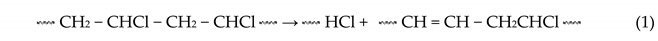



The chain in (1) may be the result of free radical ionic, or ion–radical steps. The presence of a small proportion of head-to-head, tail-to-tail, and other configurational irregularities in the backbone structure of poly(vinyl chloride) leads to more complex elimination steps by thermal degradation (alone or in the presence of catalysts such as aluminium chloride). The introduction of ring structures, a major process during dehydrochlorination, is likewise affected by the distribution of the chlorine atoms along the polymeric backbone [35].

The thermal ageing of polymers is associated with molecular deterioration due to overheating caused by tertiary hydrogen atoms in the polymer chain. The elevated temperature causes chain scission of the long chain backbone of the polymer that reacts with one another. This leads to changes in properties such as a reduction in ductility, colour change and the appearance of microcracks [36]. The thermal stability of PVC-P is related to the type and amount of plasticizer. The rate of dehydrochlorination depends on the amount of plasticizer in the PVC and is not linearly dependent [33]. PVC-P waterproofing membranes exposed to natural weathering experience a significant deterioration of the microstructure in the long term, usually due to oxidative degeneration [10].

The thermal stability of PVC-P is related to the type and amount of plasticizer. The rate of dehydrochlorination depends on the amount of plasticizer in the PVC and is not linearly dependent [34].

Examination of the microstructure by electron microscopy showed that the most surface degraded Specimen was Specimen 1, which was directly exposed to UV irradiation and high temperatures. Specimens 1 and 2 were not adequately separated from the EPS thermal insulation by polyester felt. The effect of interaction with the EPS and exposure to weathering in PVC-P results in, among other things, a significant mass reduction in chlorine and carbon. From the attached results of the EDS analysis, which are shown graphically in Figure 25, Figure 26 and Figure 27 as shown in Figure 16, Figure 20 and Figure 24, the average chlorine value as a result of the EDS analysis over the five surface area spectra is lowest in Specimen 1, averaging 21.58% by mass, and is highest in Specimen 3, averaging 38.45% by mass. Specimen 3, which was protected from direct weathering by a layer of polyester felt and gravel, has the least degradation due to dehydrochlorination, Specimen 1 has the most. A typical sign of oxidative degeneration of PVC-P is an increased oxygen level. It can be seen from Figure 28 that the average value of the oxygen element detected by spectral analysis at the five surface measurement points on Specimen 1 is the highest, averaging 16.96% by mass, which is significantly higher than that measured on Specimen 3, where the average value is 7.96% by mass. Oxidative degeneration is highest on Specimens exposed to direct weathering. The carbon element content is approximately the same in all Specimens, with a minimum of 49.65% by weight in Specimen 3 and a maximum in Specimen 2 and 3, where it averages 57.62 and 52.99% by weight, respectively. The individual results of the EDS analyses for all the chemical elements tested and for all the Specimens are given in Figure 28, Figure 29 and Figure 30.

This figure is in line with the findings of Pedrosa et al. [10], who suggest that dehydrochlorination could be the main cause of the deterioration.

The last stage of the research on PVC-P strips in relation to the deterioration of mechanical properties due to ageing was the control of the microstructure of the fracture of the Specimen under cyclic tensile loading. A piece of PVC-P waterproofing was Specimend from the dynamic tensile fracture area at the site of the mesh rupture. The Specimens were taken from Specimen 2, where the maximum value of specific strain at break was measured to be 98.78%, and from Specimen 3, where the value was 29.90%. The microstructure of the breaking surface of Specimen 2 is shown in Figure 31. The closely bonded fibres of the polyester mesh show that the degradation of the PVC-P did not progress as rigorously as in Specimen 3, even though the polymer matrix remained intact between the fibres at the point of rupture of the polyester fibres and the fracture of the material is tough, which also implies a higher rupture force of 1275 N. The degradation of the PVC-P was not as severe as in Specimen 3, but the degradation of the polyester matrix at the point of rupture of the polyester fibres was more severe.

Figure 32 shows the microstructure of the brittle fracture of the composite due to the effect of cyclic tensile loading in Specimen 3. The brittle fracture is due to the degraded polymer matrix, which no longer interacted with the reinforcement—the polyester mesh—during the failure (rupture) phase of the Specimen. The measured value of the specific strain at failure of this Specimen is 29.90% and the breaking force is 1100.19 N.

## 4. Conclusions

The mechanical performance of the waterproofing used for installation in flat roofs is a very important material characteristic based on which the vast majority of investors choose the optimal solution. As we pointed out in the introduction, deformation and breaking force are a priority for waterproofing membranes, in addition to price. This criterion can be misused by contractors when offering a suitable material if they do not take into account the experience and research associated with the lessons learned from practice when selecting the waterproofing offered.

In this paper, we aim to show that mechanical dynamic testing, which is not standardised compared to static tensile testing, can be used to predict the quality of composite materials—PVC-P waterproofing. The most relevant criterion to ensure the use value of a PVC-P waterproofing membrane is the comparison of the values of the breaking force and the fracture shape of the Specimen during the warranty period of the installed material. These properties can be easily checked before the expiry of the warranty period. Brittle fracture and reduction in ductility are certainly undesirable properties.

Our research has shown that tensile testing, which is not required by the standards, can also increase some of the declared properties. These properties can also be misleading in the assessment of the use value. We have found that the increase in elongation at break of aged PVC-P waterproofing membranes may be due to the ageing of the polymer matrix and the increase in its stiffness, which—in cooperation with the curing phase, which is a polyester mesh—may increase the elongation but reduce the breaking force.

The microstructural changes in the aged Specimens of the PVC-P waterproofing membrane exposed to weathering and ageing resulted in a reduction in the ductility of the polymer matrix, while the load-bearing role was completely taken over by the reinforcement, which is the polyester mesh. The composite waterproofing membrane has thus acquired a higher degree of deformation and the load-bearing capacity, represented by the breaking force being reduced. The reduction in the ductility of the polymer matrix in PVC-P waterproofing membranes implies a higher degree of degradation, since the composite, due to the degenerative changes in the material and the effect of wind and temperature loading, develops cracks, which cause a reduction in the primary function, which is watertightness.

UV irradiation and exposure to temperature changes on PVC-P waterproofing cause changes in the waterproofing and oxidation processes are associated with the loss of mass. Volume changes cause shrinkage which can cause further damage. These changes are reflected in a reduction in the carbon element content. Contact with EPS or XPS thermal insulation is detrimental to PVC-P waterproofing because the modest (too thin) design of the separating layer, which is polyester felt, may cause degenerative changes, which are evident by a reduction in the chlorine element content, which may result from dehydrochlorination and exposure to high temperatures. Interaction with materials such as XPS and EPS also increases the oxygen element content, which is a sure sign of oxidative degeneration. This effect is not yet well understood, but it is an important guideline for the design of flat unventilated roofs to be careful in the selection of appropriate materials.

## Figures and Tables

**Figure 1 polymers-14-01312-f001:**
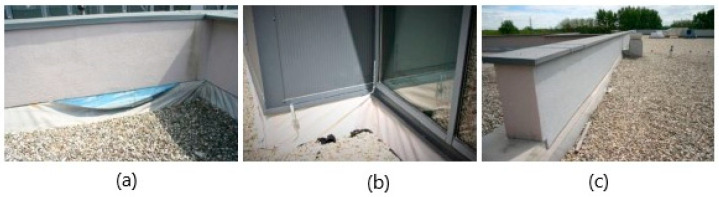
Photos of typical injuries: (**a**) damage to the detail of the waterproofing finish—tearing out of the finish below the façade attic; (**b**) the waterproofing wrinkling phenomenon; (**c**) the waterproofing membrane stretched and no longer vertically terminated at the attic.

**Figure 2 polymers-14-01312-f002:**
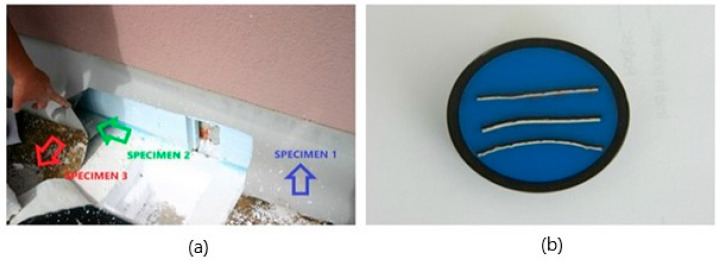
Demonstration of the PVC-P membrane sampling at the facility (**a**); the Specimen produced for analysis on the stereo optical microscope (**b**).

**Figure 3 polymers-14-01312-f003:**
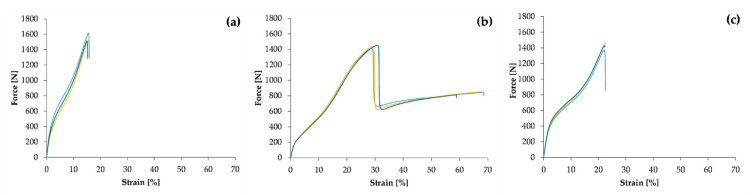
Representative tensile test curves for Specimens 1 (**a**), 2 (**b**) and 3 (**c**).

**Figure 4 polymers-14-01312-f004:**
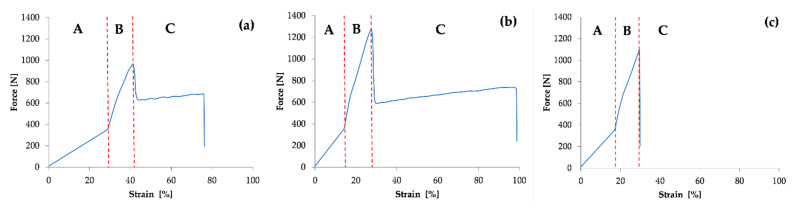
Working curves of dynamic tensile test for Specimens 1 (**a**), 2 (**b**) and 3 (**c**).

**Figure 5 polymers-14-01312-f005:**
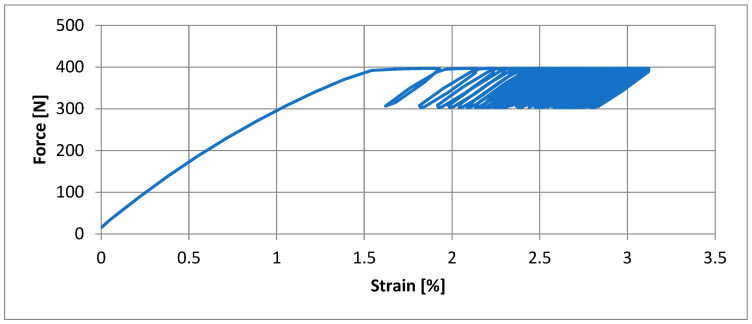
Typical segment from a cycling diagram.

**Figure 6 polymers-14-01312-f006:**
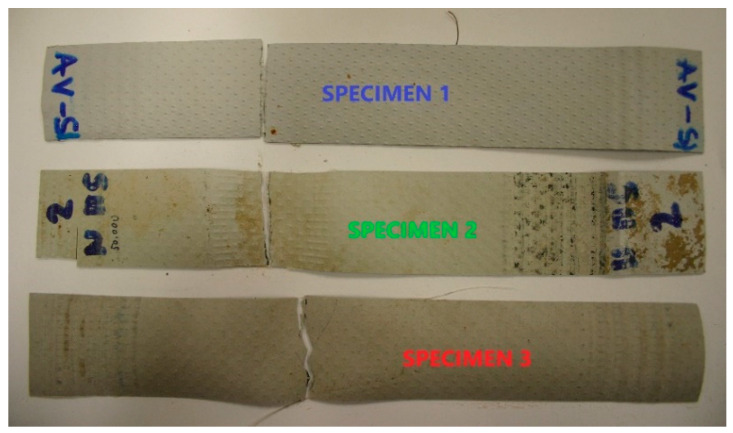
Specimens after dynamic tensile test.

**Figure 7 polymers-14-01312-f007:**
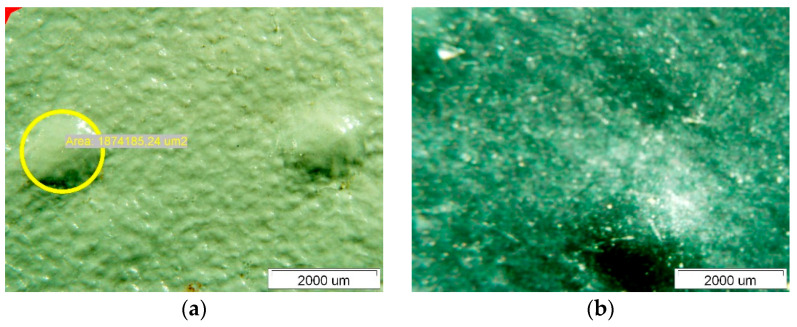
Surface of Specimen 1: (**a**) upper side, (**b**) lower side.

**Figure 8 polymers-14-01312-f008:**
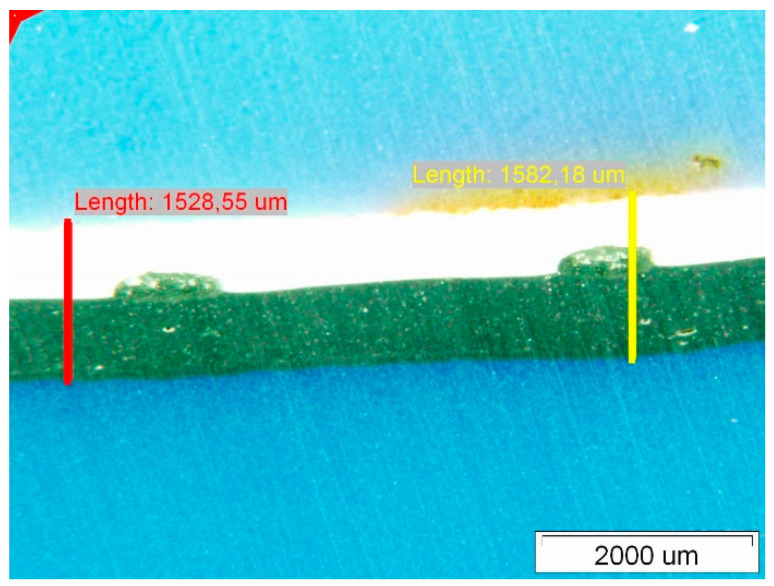
Membrane thickness measurements.

**Figure 9 polymers-14-01312-f009:**
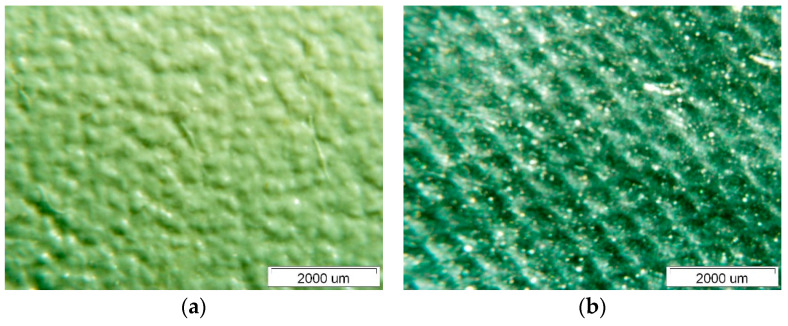
Surface of Specimen 2: (**a**) upper side, (**b**) lower side.

**Figure 10 polymers-14-01312-f010:**
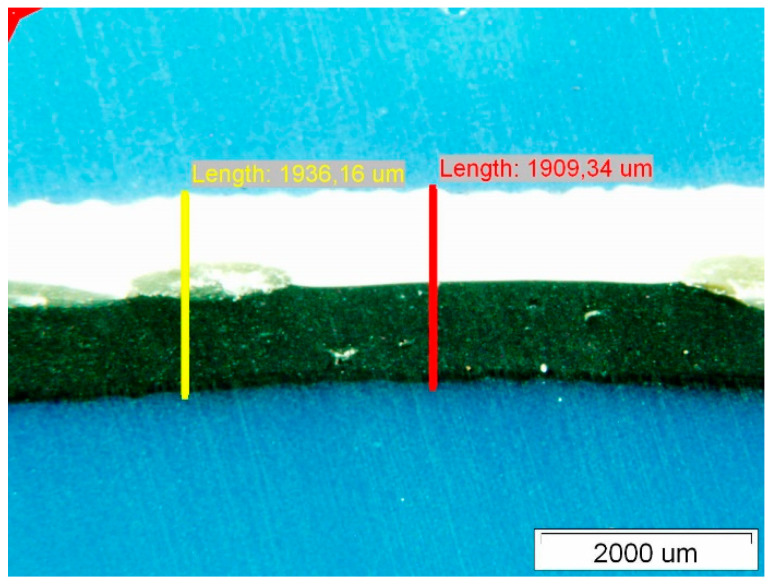
Membrane thickness measurements.

**Figure 11 polymers-14-01312-f011:**
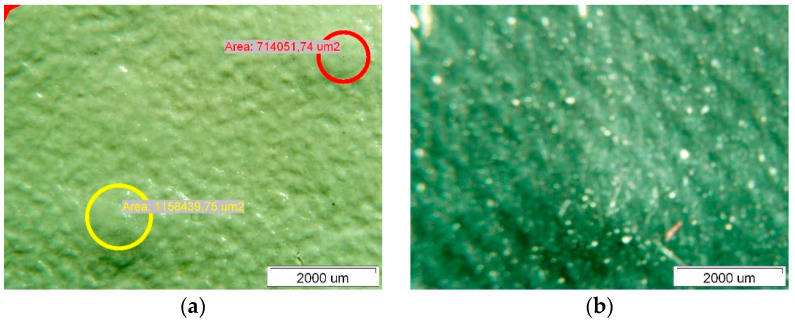
Surface of Specimen 3: (**a**) upper side, (**b**) lower side.

**Figure 12 polymers-14-01312-f012:**
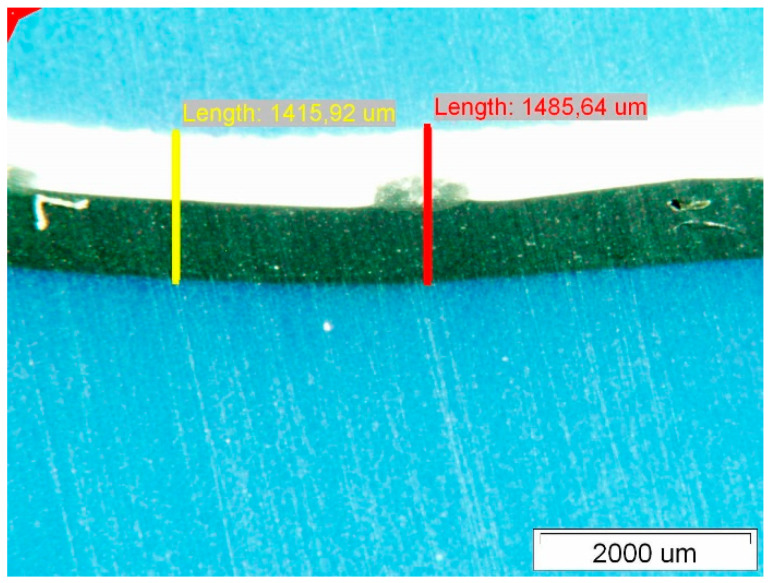
Membrane thickness measurements.

**Figure 13 polymers-14-01312-f013:**
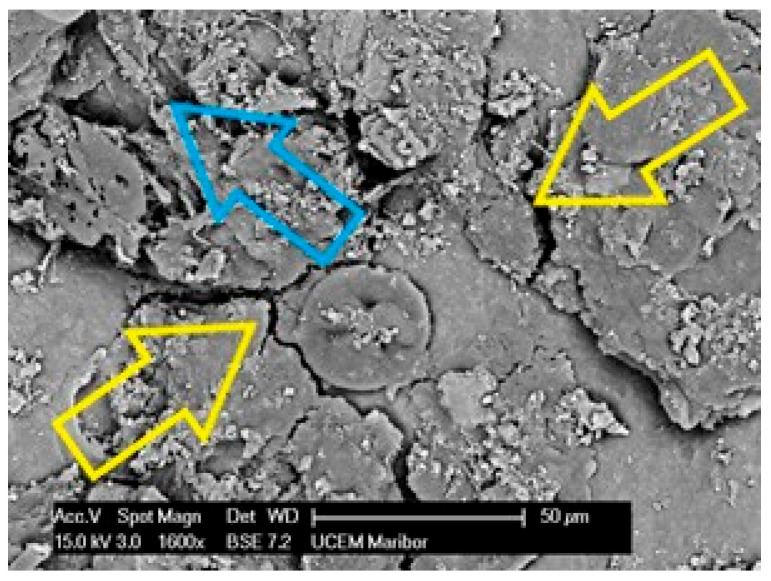
Microstructure of the surface of Specimen 1 at 1600× magnification.

**Figure 14 polymers-14-01312-f014:**
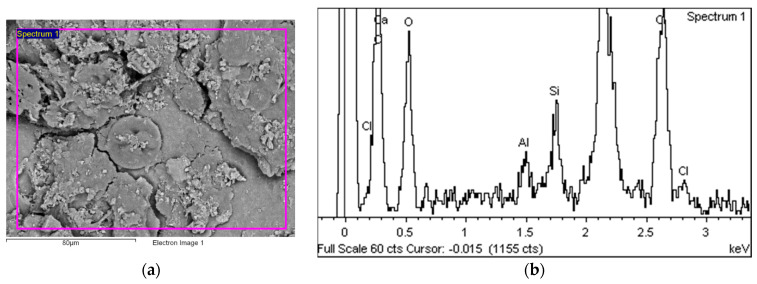
EDS analysis area (**a**) and EDS spectrum (**b**).

**Figure 15 polymers-14-01312-f015:**
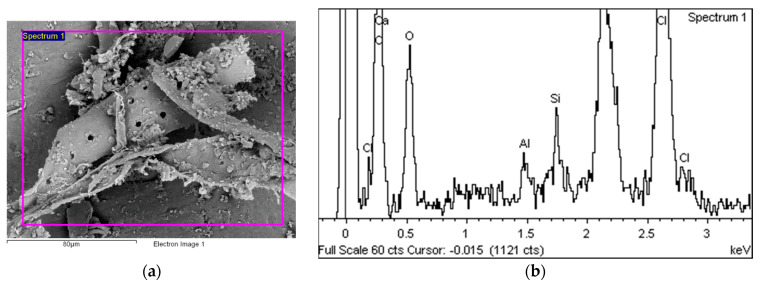
EDS analysis area (**a**) and EDS spectrum (**b**).

**Figure 16 polymers-14-01312-f016:**
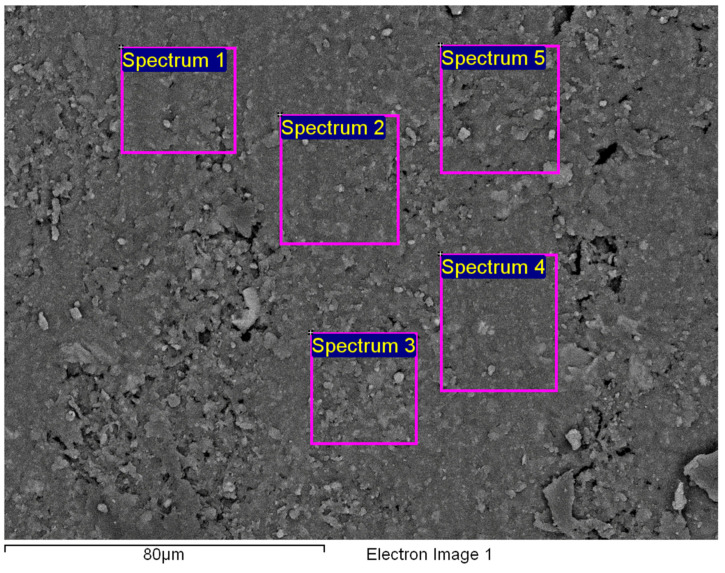
Five areas where the EDS analysis was carried out.

**Figure 17 polymers-14-01312-f017:**
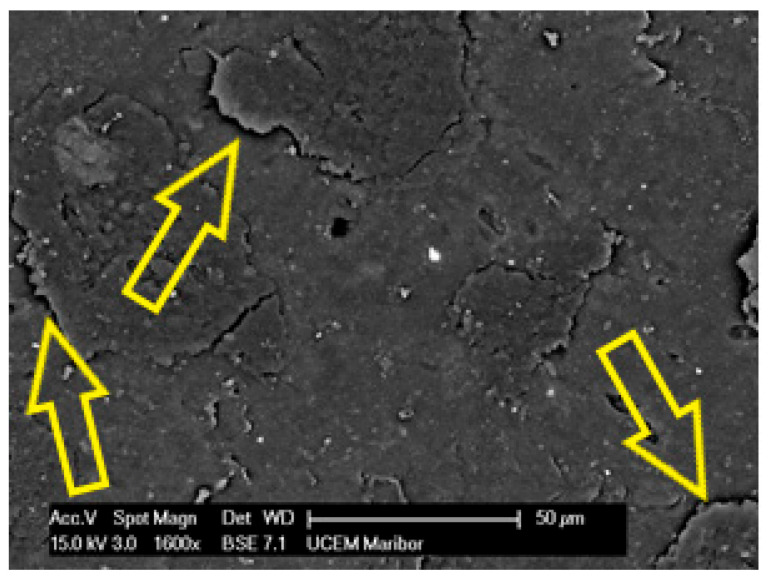
Microstructure of the surface of Specimen 2 at 1600× magnification.

**Figure 18 polymers-14-01312-f018:**
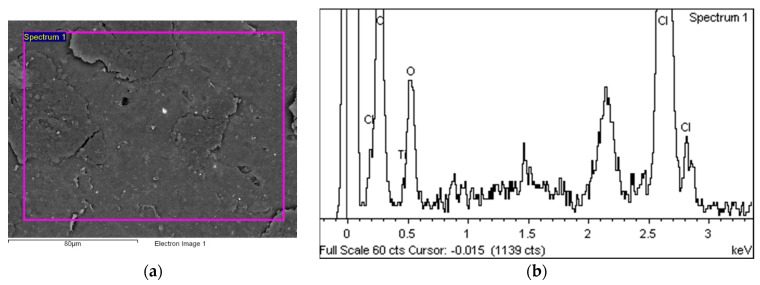
EDS analysis area (**a**) and EDS spectrum (**b**).

**Figure 19 polymers-14-01312-f019:**
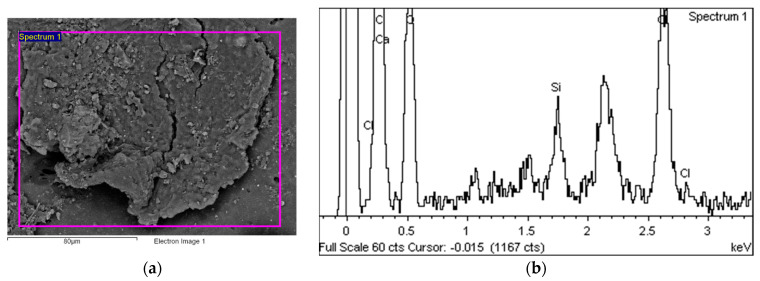
EDS analysis area (**a**) and EDS spectrum (**b**).

**Figure 20 polymers-14-01312-f020:**
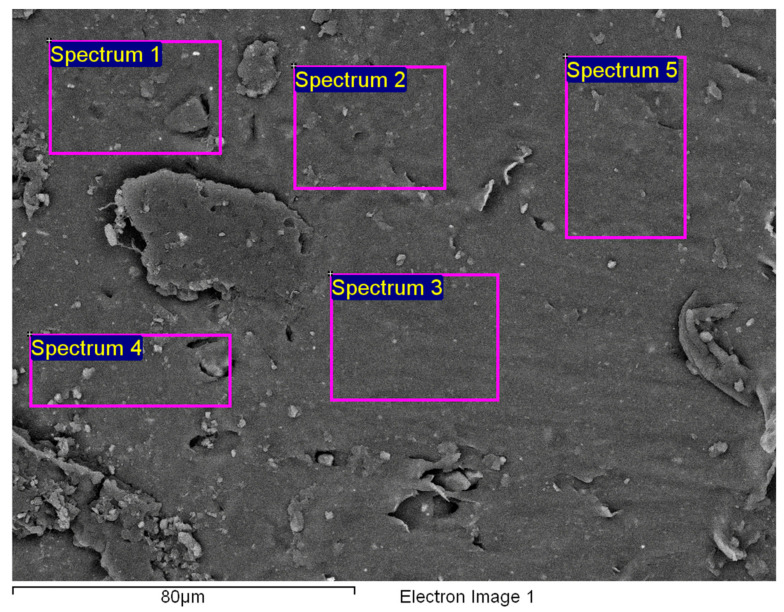
Five areas where the EDS analysis was carried out.

**Figure 21 polymers-14-01312-f021:**
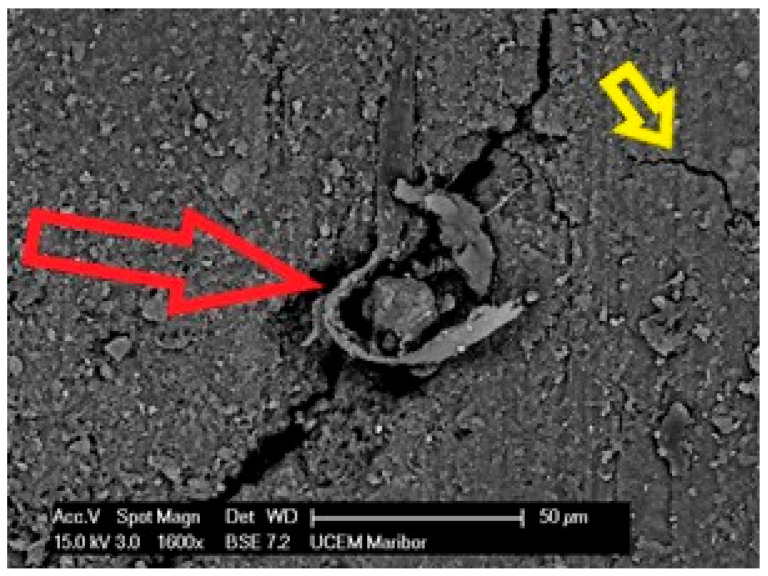
Top surface of Specimen 3 at 1600× magnification.

**Figure 22 polymers-14-01312-f022:**
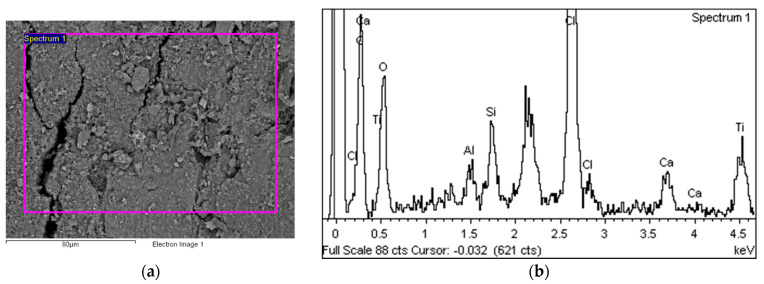
EDS analysis area (**a**) and EDS spectrum (**b**).

**Figure 23 polymers-14-01312-f023:**
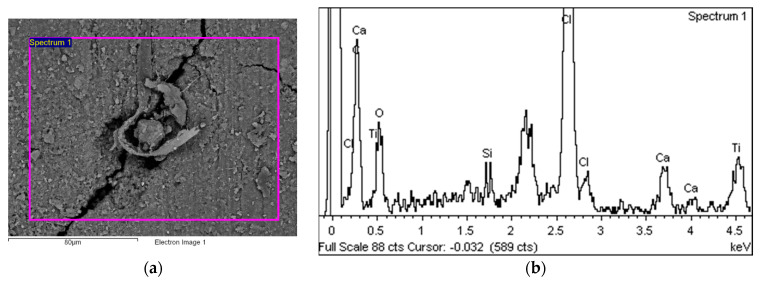
EDS analysis area (**a**) and EDS spectrum (**b**).

**Figure 24 polymers-14-01312-f024:**
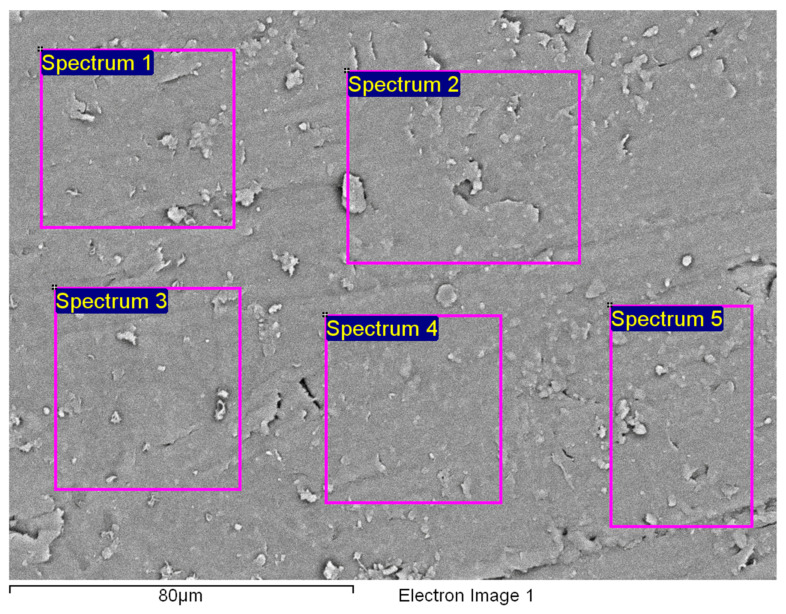
Five areas where the EDS analysis was carried out.

**Figure 25 polymers-14-01312-f025:**
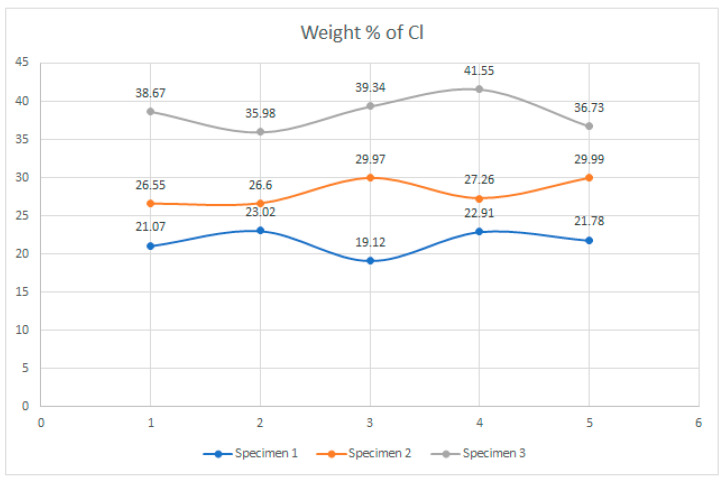
Result of EDS analysis, presence of an element, Cl, in mass percentage.

**Figure 26 polymers-14-01312-f026:**
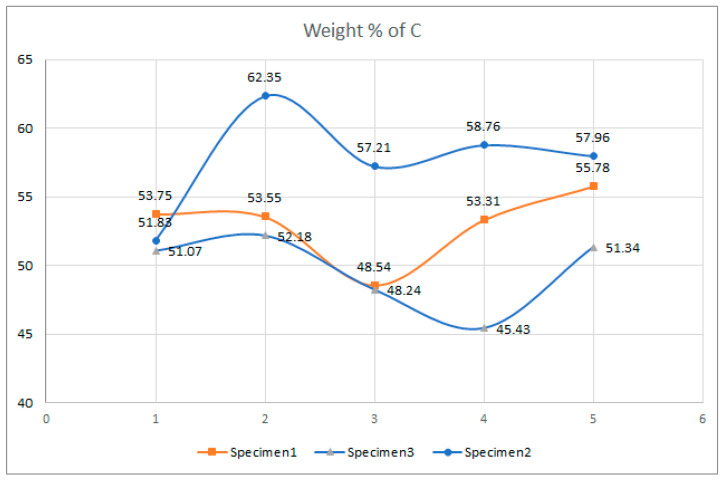
Result of EDS analysis, presence of an element, C, in mass percentage.

**Figure 27 polymers-14-01312-f027:**
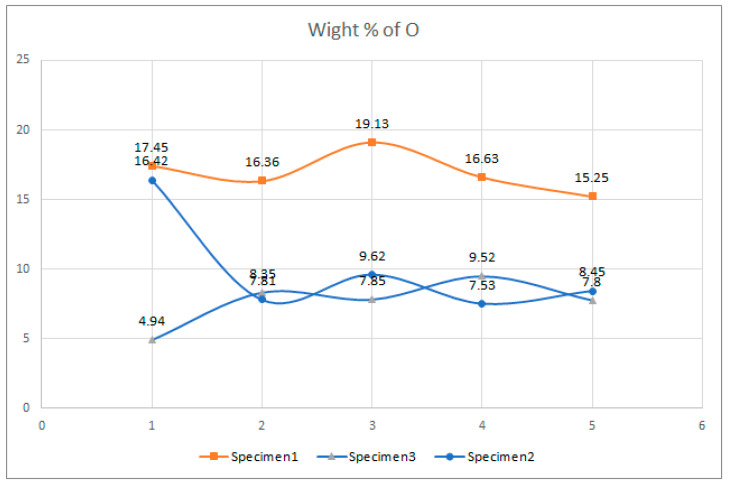
Result of EDS analysis, presence of an element, O, in mass percentage.

**Figure 28 polymers-14-01312-f028:**
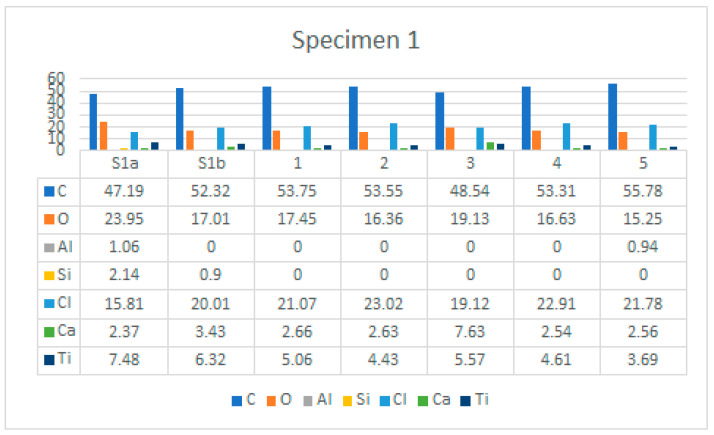
EDS analysis results for Specimen 1, for all areas analysed.

**Figure 29 polymers-14-01312-f029:**
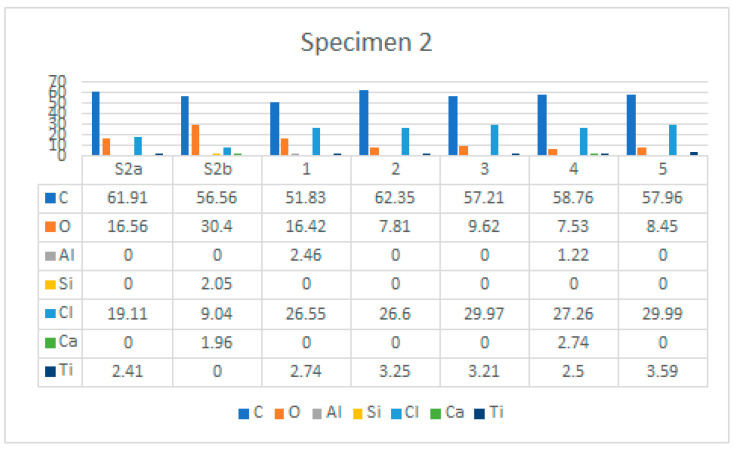
EDS analysis results for Specimen 2, for all areas analysed.

**Figure 30 polymers-14-01312-f030:**
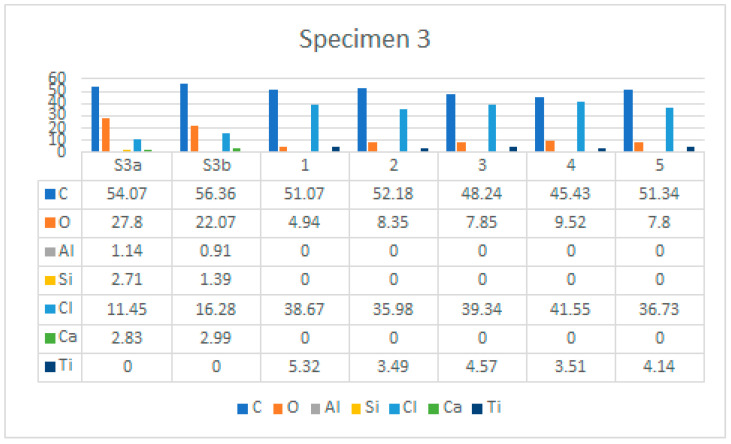
EDS analysis results for Specimen 3, for all areas analysed.

**Figure 31 polymers-14-01312-f031:**
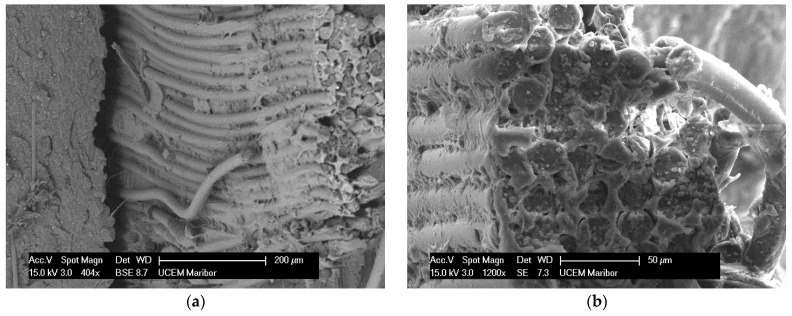
The micrograph shows a tough fracture of a composite membrane. The fibres of the cured phase are bonded to the PVC-P matrix (**b**) and there is a clear slip at the point where the fibres are pulled out of the matrix (**a**).

**Figure 32 polymers-14-01312-f032:**
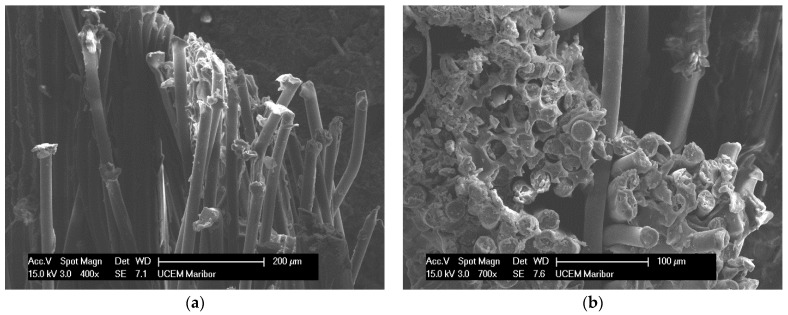
The microphotographs show a brittle fracture of the Specimen 3 without the formation of a neck in tensile test. The fibres have been pulled out of the PVC-P matrix and are no longer connected to each other (**a**,**b**).

**Table 1 polymers-14-01312-t001:** Results of mechanical properties for Specimens 1, 2 and 3.

Series	Specimen	F_max_	ε_Fmax_	ε_Break_	E_S_^350^
1	1	1612.38	15.56	15.86	
	2	1548.32	15.99	16.30	
	3	1514.78	15.00	15.29	
	x *	1558.49	15.51	15.82	3.55
	s **	49.59	0.49	0.51	
2	1	1412.70	28.74	68.62	
	2	1439.88	29.41	68.06	
	3	1449.29	30.77	58.89	
	x *	1433.96	29.64	65.19	1.12
	s **	19.0	1.04	5.46	
3	1	1374.42	21.94	22.67	
	2	1455.11	22.42	22.84	
	3	1430.01	22.11	22.27	
	x *	1419.85	22.16	22.60	3.43
	s **	41.29	0.25	0.29	

*—average; **—standard deviation; E_S_^350^—secant modulus of elasticity at force 350 N.

**Table 2 polymers-14-01312-t002:** Results of mechanical properties for Specimens 1, 2 and 3.

Specimen	F_max_ [N]	ε_A_ [%] ^1^	ε_B_ [%] ^2^	ε_C_ [%] ^3^
Specimen 1	966.84	29.16	40.88	76.28
Specimen 2	1275.00	14.28	27.75	98.78
Specimen 3	1100.19	17.02	29.38	29.90

^1^—strain at the end of conditioning; ^2^—strain at F_max_; ^3^—strain at break.

**Table 3 polymers-14-01312-t003:** Results of optical microscope thickness measurements on Specimens.

Specimen	t_max_ [μm]	t_min_ [μm]	x [μm]	s
Specimen 1	1651.9	1448.1	1533.02	73.49
Specimen 2	1936.16	1844.98	1903.98	40.85
Specimen 3	1689.45	1415.92	1509.39	88.50

t—thickness; x—average thickness; s—standard deviation.

**Table 4 polymers-14-01312-t004:** EDS analysis results for Specimen 1.

Element	S1a	S1b	1	2	3	4	5
C	47.19	52.32	53.75	53.55	48.54	53.31	55.78
O	23.95	17.01	17.45	16.36	19.13	16.63	15.25
Al	1.06	0	0	0	0	0	0.94
Si	2.14	0.9	0	0	0	0	0
Cl	15.81	20.01	21.07	23.02	19.12	22.91	21.78
Ca	2.37	3.43	2.66	2.63	7.63	2.54	2.56
Ti	7.48	6.32	5.06	4.43	5.57	4.61	3.69

**Table 5 polymers-14-01312-t005:** EDS analysis results for Specimen 2.

Element	S2a	S2b	1	2	3	4	5
C	61.91	56.56	51.83	62.35	57.21	58.76	57.96
O	16.56	30.4	16.42	7.81	9.62	7.53	8.45
Al	0	0	2.46	0	0	1.22	0
Si	0	2.05	0	0	0	0	0
Cl	19.11	9.04	26.55	26.6	29.97	27.26	29.99
Ca	0	1.96	0	0	0	2.74	0
Ti	2.41	0	2.74	3.25	3.21	2.5	3.59

**Table 6 polymers-14-01312-t006:** EDS analysis results for Specimen 3.

Element	S3a	S3b	1	2	3	4	5
C	54.07	56.36	51.07	52.18	48.24	45.43	51.34
O	27.8	22.07	4.94	8.35	7.85	9.52	7.8
Al	1.14	0.91	0	0	0	0	0
Si	2.71	1.39	0	0	0	0	0
Cl	11.45	16.28	38.67	35.98	39.34	41.55	36.73
Ca	2.83	2.99	0	0	0	0	0
Ti	0	0	5.32	3.49	4.57	3.51	4.14

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
