# Peer review of "Durability and Degradation of PVC-P Roofing Membrane—Example of Dynamic Fatigue Testing"

_polymers, 2022, doi:10.3390/polym14071312_

Round 1

Reviewer 1 Report

The current MS investigated waterproofing of PVC-P membrane after a service life of 11 years. The work is interesting and would be applicable and useful from the industrial and R&D sectors' points of view. However, some comments are recommended to address in order to improve the MS.
1. The introduction slightly needs to be modified to clearly demonstrate the essentials of the work by adding more references, explaining the advantages of the PVC, their wide range of applications and what has been done in brief and etc.
2. In some parts of the MS, what is claimed needs to be cited, for instance: lines 42, 57, 95, 102,113, and the rest need to be checked by the authors. The following papers need to be cited as described the hydrophilicity behavior of the polymeric membranes. https://pubs.rsc.org/en/content/articlehtml/2020/ra/d0ra07592b.

3. How the cross-section of the samples was changed during this study.

4.  It is suggested to clearly describe the degradation mechanism of the samples.

Author Response

Dear Reviewer 1

Thank you for your valuable comments and suggestions for corrections.

We have corrected the introduction and increased the number of citations - highlighted in red.

We have added explanatory notes in the text along lines 42, 57, 95, 102 and 113, also highlighted in red.

We have corrected the thickness of the samples and added text referring to the mechanism of degradation of PVC.

Thank you again and best regards.

Lubej & A. Ivanič

Reviewer 2 Report

This manuscript isnot a complete scientific research paper. In this manuscript, the fatigue testing of PVC-P roofing membrane have done, and the testing results were presented only in this manucript necessary discussion and analysis. The reasons that led to the degradation of PVC-P roofing membrane, and the degradation mechanism should be discussed in the manuscript.  It is necessary to do comparative experiments between the new and fatigue PVC-P roofing membrane, by which difference test results between the  new and fatigue PVC-P roofing membrane should be discussed. 

Author Response

Dear Reviewer 2

Thank you for your valuable comments and suggestions for corrections.

We have corrected the introduction and increased the number of citations - highlighted in red.

We have not carried out any investigations on unstained samples because we do not have the samples. However, we agree with you that the study would be more comprehensive if we had them.

Best regards

Lubej & A. Ivanič

Reviewer 3 Report

  1. Add more introduction section and compare the previous studies with their merits and demerits.
  2. Provide high-quality images Fig. 3, 4, 7,8,9,10.
  3. In SEM images high light the characteristics and elements.
  4. Provide error of analysis and error bars in Fig. 25 to 27.
  5. What about the sensitivity and uncertainty of results.

Author Response

Dear Reviewer 3

Thank you for your valuable comments and suggestions for corrections.

We have corrected the introduction and increased the number of citations - highlighted in red.

We have corrected the quality of the images and added an analysis of the chemical elements present (error bars) to the chapter.Thank you again and best regards.

Lubej & A. Ivanič

Round 2

Reviewer 2 Report

This is suitable for publication.